# Redox-innocent scandium(III) as the sole catalyst in visible light photooxidations

Amal Hassan Tolba [1,2] ✉, Ahmed M. El-Zohry [3] ✉, Jafar Iqbal Khan [4], Eva Svobodová[1], Josef Chudoba[5], Jiří Klíma[6], Karol Lušpai [6,7], Martin Pižl [8], Jiří Šturala [8] & Radek Cibulka [1] ✉

In recent years, the catalytic activity of scandium triflate $Sc(OTf)_3$ has attracted significant attention due to its robust Lewis acidity and the oxophilicity of $Sc^{3+}$. These features have led to impressive progress in developing diverse organic reactions, including C-C bond formation. The $Sc^{3+}$ also facilitates single electron transfer in photoinduced reactions either by coordination to an organophotoredox catalyst, which modifies its redox reactivity, or by the formation of a scandium–superoxide anion complex after electron transfer from a light-absorbing redox-active compound. The prior consideration of $Sc^{3+}$ as a redox-inactive/innocent metal ion initially hampered the investigation of the possibility of using $Sc(OTf)_3$ as a sole visible light photoredox catalyst. This work demonstrates the use of $Sc(OTf)_3$ as a visible light photocatalyst capable of direct and mild aerobic oxidative C-H functionalisation of aromatic substrates by oxidation of the benzylic position and direct cyanation of the aromatic ring.

Rare-earth metal ions (Sc, Y, lanthanides) have attracted significant attention, mainly due to their Lewis acidity and oxophilicity. In particular, $Sc(OTf)_3$ has been introduced in recent decades as a promising Lewis acid for C-H functionalisation and C-C bond formation reactions[1,2]. In the past 20 years, scandium salts have also effectively assumed a role in photoredox catalysis—a process that has emerged as a full-fledged alternative in organic synthesis, often unlocking transformations that are not attainable in the dark[3–6]. Photoredox catalysis relies on an electron transfer process between a highly reactive, light-excited catalyst and a redox-labile substrate (S). The result is a reactive radical intermediate that undergoes further transformations, ultimately resulting in the formation of new bonds under mild conditions.

Photoredox catalysis requires a catalyst that is both photochemically and redox active—typically a compound derived from an organic dye or a complex of a redox-active metal, such as ruthenium or iridium (see Fig. 1a)[7–10]. Conversely, redox-inactive/innocent metals, such as scandium, which have a predominantly trivalent redox-stable state, are not directly involved in photoredox reactions[11–13]. Nevertheless, these metals, and particularly $Sc(OTf)_3$, can serve as Lewis acids (LA) that activate substrates/organophotocatalysts by altering their redox characteristics, thereby facilitating their participation in photoredox and photosensitisation reactions (Figs. 1b and 1c)[12–18]. For instance, the redox properties of a flavin derivative (Fl) were significantly enhanced via their coordination to $Sc(OTf)_3$. This alteration enabled the oxidation of some electron-deficient substrates via a SET process between the substrate and flavin in excited $Fl\text{-}2Sc^{3+}$ complex (Fig. 1c, right); such reaction cannot be achieved without $Sc(OTf)_3$[19,20].

Notably, the oxophilicity of some $M^{n+}$ cations facilitates their binding to superoxide radical anion $(O_2^{\cdot-})$ to form superoxide–metal

[1]Department of Organic Chemistry, University of Chemistry and Technology, Prague, Prague 166 28, Czech Republic. [2]Chemistry Department, Faculty of Science, Assiut University, Assiut 71515, Egypt. [3]Ultrafast Laser Spectroscopy Lab, Center for Integrative Petroleum Research, KFUPM, Dhahran 31261, Saudi Arabia. [4]Department of Physics, School of Natural Sciences, University of Hull, Cottingham Road, Hull HU6 7RX, United Kingdom. [5]Central Laboratories, University of Chemistry and Technology, Prague, Prague 166 286, Czech Republic. [6]Department of Molecular Electrochemistry and Catalysis, J. Heyrovský Institute of Physical Chemistry, AS CR, v.v.i., 182 23 Prague 8, Czech Republic. [7]Institute of Physical Chemistry and Chemical Physics, Faculty of Chemical and Food Technology, Slovak University of Technology in Bratislava, Bratislava 81237, Slovak Republic. [8]Department of Inorganic Chemistry, University of Chemistry and Technology, Prague, Prague 166 28, Czech Republic. ✉e-mail: amal.hassan44@yahoo.com; ahmed.elzohry@kfupm.edu.sa; cibulkar@vscht.cz

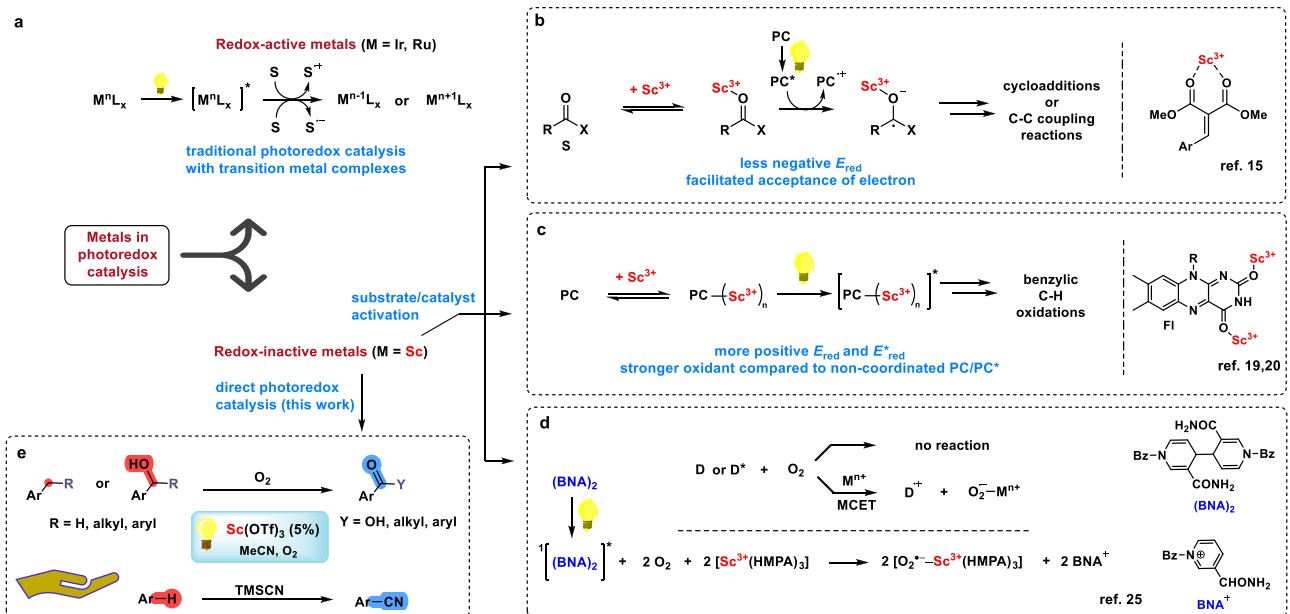

**Fig. 1 | Role of redox-active and redox-inactive metal ions in photochemical transformations. a** Use of metal ions in photoredox catalysis[3,8–10]. **b** Use of Sc³⁺ as a Lewis acid to activate a substrate (S) for reaction with excited photocatalyst (PC*); example shown on right. **c** Activation of organic photocatalyst (PC) with Sc³⁺ as Lewis acid (LA); example shown on right. **d** Metal ion-coupled electron transfer (MCET) with oxygen as an acceptor; an example involving Sc³⁺ in a procedure with light absorbing substrate is shown. **e** Use of Sc³⁺ salt as the sole catalyst in photoredox catalysis for benzylic oxidations and cyanations (this work).

complexes ($M^{n+}$–$O_2^{•-}$) that have been recognised within the reductive activation of dioxygen by metalloenzymes in numerous biological redox reactions that employ $O_2$ as an oxidant[21–24]. This process has been mimicked in the dioxygen activation cycle during photoredox transformations, commencing with the photoinduced single electron transfer (PET) from a light-absorbing, redox-active compound to $O_2$ and followed by the generation of $M^{n+}$-$O_2^{•-}$ complexes (Fig. 1d, upper part). Binding of $M^{n+}$ to oxygen enhances its ability to accept an electron. Thus, an electron transfer from donor (D) to $O_2$, which is thermodynamically infeasible in the absence of metal ions, becomes possible in the presence of $M^{n+}$. This process is generally called metal ion–coupled electron transfer (MCET)[12,25–29]. Scandium³⁺ ion, with its smallest radius among $M^{3+}$, can bind strongly with $O_2^{•-}$, making it far more effective than other $M^{n+}$ ions for providing SET reactions driven by $M^{n+}$-$O_2^{•-}$ complex formation[12,30–35].

A review of previous reports on the role of Sc(OTf)₃ in photoinduced electron transfer (PET) transformations involving the reductive activation of dioxygen confirms that a light-absorbing electron donor and additional stabilisation are necessary. For example, Fukuzumi[30] recorded the EPR spectrum of the (HMPA)₃Sc³⁺-$O_2^{•-}$ complex stabilised by three equivalents of hexamethylphosphoramide (HMPA) ligand, formed via PET from the singlet excited state of 1-benzyl-1,4-dihydronicotinamide dimer[1][(BNA)₂]*, to $O_2$. This procedure was accompanied by fast cleavage of the C-C bond in the formed (BNA)₂⁺ (Fig. 1d, bottom part).

In this work we demonstrate the activity of Sc(OTf)₃ functioning as a sole photoredox catalyst. Sc(OTf)₃ is shown to mediate aerobic oxidation of a benzylic C-H bond and direct oxidative cyanation of arenes as examples of C-H activation and C-C coupling reactions (Fig. 1e). These procedures dispel the usual notion of scandium salts as redox-inactive species useful mostly as Lewis acids and expand their possible application as simple, readily available photocatalysts. In general, photoredox processes might be a future opportunity for scandium, which is abundantly found on earth but remains an underutilised metal[36].

## Results

### Development of Sc(OTf)₃-based aerobic photooxidations

Upon the surprising observation of the photocatalytic activity of Sc(OTf)₃, we directed our attention towards the aerobic oxidation of toluene (**1a**; $E_{ox}$ = +2.36 V vs SCE[37]) as a benchmark. The oxidation was performed with 5 mol% of Sc(OTf)₃ ($E_{red}$ < −1.0 V vs SCE, see ref. 38 and our experiments below; i.e. not strong enough in ground state to oxidise **1a**) in $O_2$–saturated acetonitrile (MeCN) under blue light irradiation (400 nm LED) for 24 h. Encouragingly, this experiment afforded benzoic acid (**2a**) as the sole product at an 80% yield (Fig. 2a, entry 1). Under anaerobic conditions, or in the absence of light or Sc(OTf)₃, the reaction did not proceed (entries 2–5). This system demonstrated significantly decreased effectiveness with 450 nm LED illumination (Entry 6). An assessment of Lewis acids revealed that Sc(OTf)₃ was the optimal selection, probably due to its superior ability to bind strongly with $O_2^{•-}$ [35]. By contrast, employing Mg(OTf)₂, Zn(OTf)₂, La(OTf)₃, and Ba(OTf)₂ under identical experimental conditions resulted in the complete cessation of the reaction (see Supplementary Section S2).

As depicted in Fig. 2b, the choice of solvent was pivotal in determining the reaction progression. A switch from nitrile solvents, such as MeCN or EtCN, to MeNO₂ resulted in some product, but at a decreased yield compared to nitrile solvents. A mixture of MeCN and MeOH (9:1) resulted in a significant decrease in the reaction yield, with complete cessation in MeOH or water, as examples of protic solvents. The use of polar aprotic solvent (DMSO or DMF) or nonpolar solvents also resulted in complete inhibition of the reaction.

An investigation of catalyst loading (Fig. 2a, entries 7 and 8 and Supplementary Section S2) also revealed that using a smaller amount of Sc(OTf)₃, at 2.5 mol%, slightly reduced the yield of benzoic acid (**2a**), whereas employing 10 mol% of the catalyst notably achieved a full conversion to **2a**. Moreover, the time-dependent yield profile showed an almost constant rate of toluene (**1a**) oxidation with increasing reaction time (Fig. 2c). Consequently, the plot of turnover frequency (TOF) values over the reaction time (4–20 h) remained stable, signifying maintenance of the photocatalytic activity of Sc(OTf)₃ (see Supplementary Section S2).

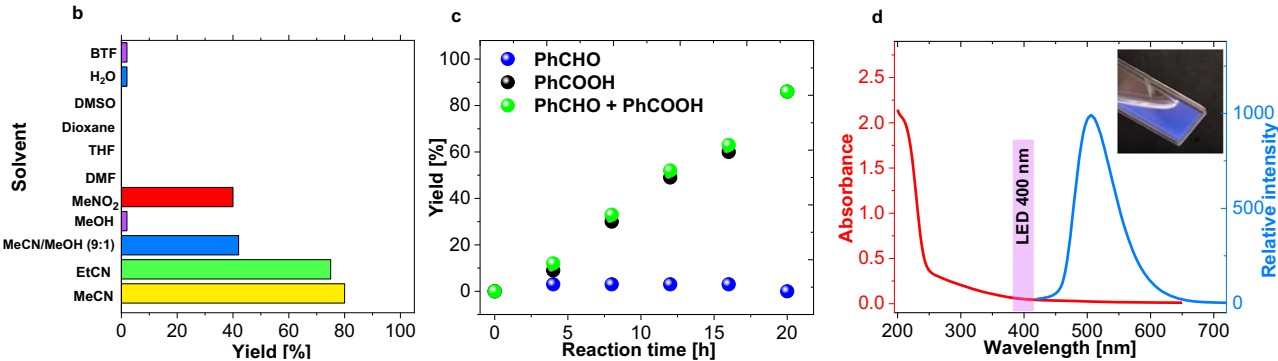

| Entry | Deviations from Optimized Conditions[a] | Yield[b] [%] |
|---|---|---|
| 1 | None | 80 |
| 2 | No Sc(OTf)$_3$ | n.d.[c] |
| 3 | No light | n.d.[c] |
| 4 | Air instead of O$_2$ | n.d.[c] |
| 5 | Argon instead of O$_2$ | n.d.[c] |
| 6 | 450 nm instead of 400 nm | traces |
| 7 | 2.5 mol% instead of 5 mol% | 72 |
| 8 | 10 mol% instead of 5 mol% | quant. |

**Fig. 2 | Sc(OTf)$_3$ can act as a sole photocatalyst. a** Control experiments showing Sc(OTf)$_3$-based aerobic photocatalytic oxidation of toluene. **b** Screening of solvents. **c** Time-dependent NMR yield profile **d** UV-Vis absorption and fluorescence spectra ($\lambda_{ex}$ = 400 nm) of Sc(OTf)$_3$ in MeCN ($c$ = 0.05 M) and a range of LED emission; an image of the solution in a 1 cm cuvette (inset). [a] Standard reaction conditions: toluene (0.140 mmol), Sc(OTf)$_3$ (5 mol%), MeCN (0.25 mL), 45 °C, O$_2$ (balloon), blue LED (400 nm), and 24 h. [b] Yields were determined from the $^1$H NMR spectra. [c] Product was not detected.

The unexpected finding of the photocatalytic activity of Sc(OTf)$_3$ prompted us to check our experiments thoroughly multiple times. We ensured that no organic impurities were driving the reaction. We also tested Sc(OTf)$_3$ from different suppliers, but we found no significant variations except for reduced activity caused by traces of oxygen atom–containing solvents like MeOH, attributable to synthesis residues. This reduced activity could be improved by recrystallising Sc(OTf)$_3$ from its solution in MeCN and using chloroform for precipitation.

The photocatalytic activity could be ascribed to the weak absorption of Sc(OTf)$_3$ in MeCN (this extended to the visible region around 400 nm), and it was also indicated by blue fluorescence with a distinct band at 505 nm observed upon excitation with 400 nm light (Fig. 2d).

### Substrate scope of Sc(OTf)$_3$-based aerobic photooxidation

Using our optimised conditions, we evaluated various substrates. Both electron-donating and electron-deficient aromatic compounds (Fig. 3a) revealed the promising photocatalytic activity of Sc(OTf)$_3$ in most cases. Toluene (**1a**) yielded 80% benzoic acid (**2a**) and *p*-chlorotoluene (**1b**) ($E_{ox}$ = +2.21 V vs SCE[37]) underwent even complete conversion to its respective carboxylic acid **2b**. As anticipated, the electron-deficient 4-(trifluoromethyl)toluene (**1c**), with the highest oxidation potential ($E_{ox}$ = +2.61 V vs SCE[37]), displayed considerable resistance to oxidation, resulting in trace product formation. Conversely, the less electron-deficient ester group-containing derivative **1 d** ($E_{ox}$ = +2.45 V vs SCE[39]) was effectively oxidised. Substrates with

higher oxidation potentials than 2.5 V vs SCE seem beyond the capabilities of Sc(OTf)$_3$–catalysed oxidation. However, we observed that combination of high temperature (65 °C) and the addition of 0.2 equivalent of trifluoroacetic acid allows the oxidation of even such a demanding substrate as 4-(trifluoromethyl)toluene (**1c**) (see Fig. 3). It should be noted that stronger trifluoromethanesulfonic acid or the weaker acetic acid did not cause such a significant increase in oxidation conversion (see Supplementary Section S2 for details).

Unexpectedly, 4-methylanisole **1e** ($E_{ox}$ = +1.51 V vs SCE[37]), despite being the most electron-rich substrate, gave only a 4% yield. This could be attributed to the back-electron transfer (BET) process during the oxidation of **1e**. Biphenyl **3** ($E_{ox}$ = +1.95 V vs SCE[40]) underwent efficient oxidation, providing a complete yield of benzoic acid (**2a**). Oxidation of *p*-xylene (**1 f**) afforded a mixture of 4-methylbenzoic acid (**2 f**) (67%) and terephthalic acid (**2 g**) (33%) after 24 h of irradiation. Ethylbenzene derivatives **4a-c** provided reasonable yields of benzoic acid derivatives according to the electronic effect, although 4-ethyltoluene (**4 d**) was oxidised to a mixture of 4-acetylbenzoic acid (**2 h**) and **2 f**. By contrast, cumene (**5**) gave a poor yield of benzoic acid upon oxidation. Diphenylmethane (**6**) was transformed into a mixture of the corresponding ketone **7** and benzoic acid (**2a**). 9H-fluorene (**8**) was converted into fluorenone (**9**) as a sole product with excellent yield, while stilbene (**10**) yielded benzoic acid via oxidative C = C cleavage.

Further investigation of the oxidation of various primary and secondary aryl alcohols, as depicted in Fig. 3b, revealed that all the tested primary benzyl alcohols, including electron-rich and highly electron-deficient ones **11a-d** and *o*-chlorobenzyl alcohol (**11e**), were

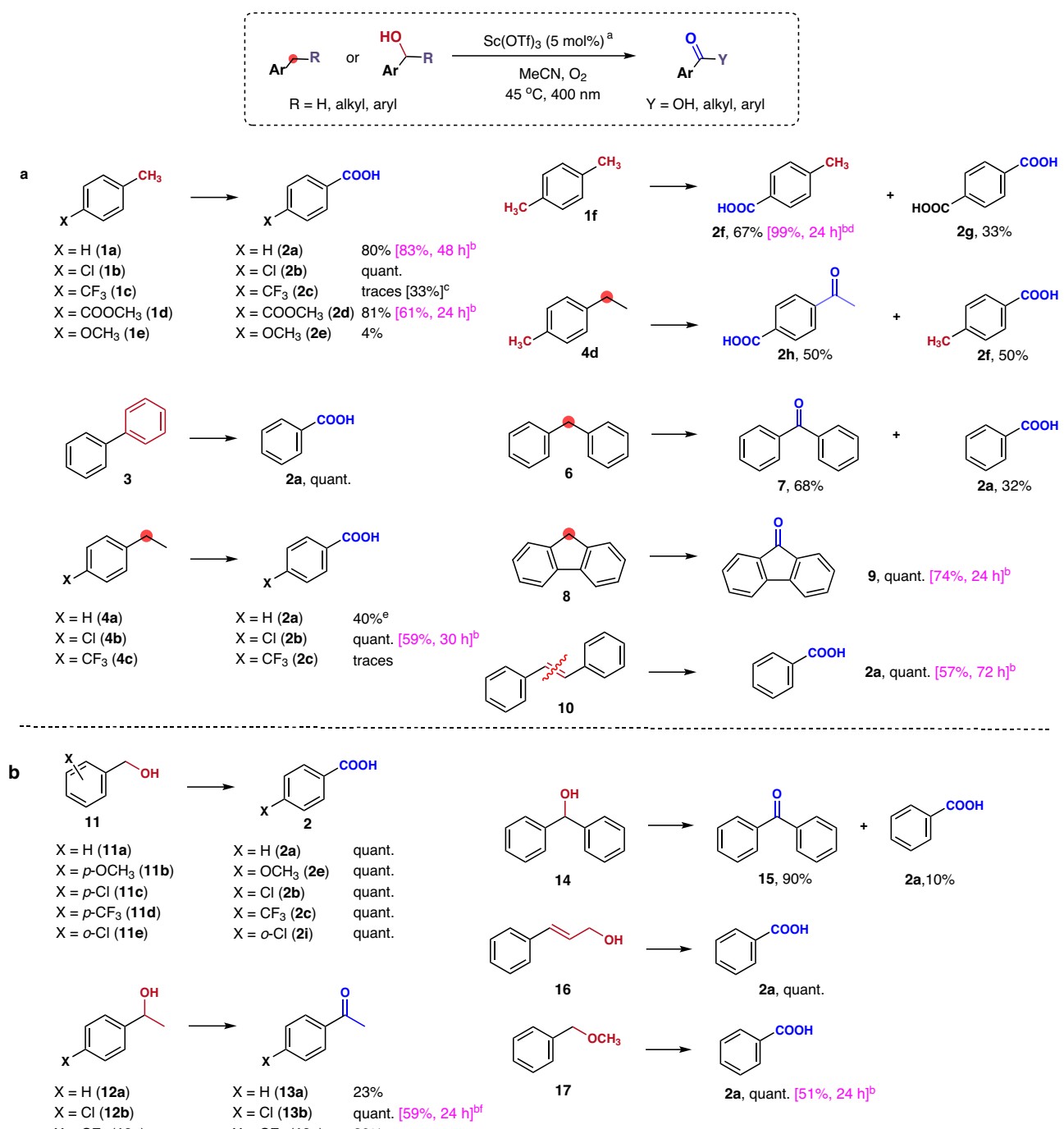

**Fig. 3 | Scope of Sc(OTf)₃-based aerobic photocatalytic oxidations. a** Benzylic oxygenations/oxidations of alkyl/alkenyl/aryl arenes. **b** Oxidations of primary and secondary benzyl alcohols and related substrates. ᵃ Standard reaction conditions: substrate (0.140 mmol), Sc(OTf)₃ (5 mol%), MeCN (0.25 mL), 45 °C, O₂ (balloon), blue LED (400 nm), and 24 h. Yields were determined by ¹H NMR spectra unless otherwise stated. ᵇ Isolated yield (in magenta); experiments were performed at a 1 mmol scale (see Supplementary Section S3 for details). ᶜ in propionitrile at 65 °C with CF₃COOH (0.2 equiv.), 48 h. ᵈ Only **2 f** as the sole product. ᵉ 48 h. ᶠ 59% of **13b** and 16% of **2b**, 24 h.

oxidised to the corresponding benzoic acids **2** in quantitative yields. The secondary alcohol 1-(4-chlorophenyl)ethan-1-ol (**12b**) provided 4-chloroacetophenone (**13b**) in a quantitative yield, while the electron-poor 1-[4-(trifluoromethyl)phenyl]ethanol (**12c**) produced a low yield of 4-(trifluoromethyl)acetophenone (**13c**), as anticipated. Surprisingly, the oxidation of 1-phenylethanol (**12a**) resulted in a low conversion to the corresponding ketone **13a**, for a yield of only 23%. A mixture of benzophenone (**15**) and benzoic acid was obtained following the oxidation of diphenylmethanol (**14**). By contrast, cinnamyl alcohol (**16**)

underwent a C=C bond cleavage during oxidation to generate a quantitative yield of benzoic acid. Notably, 4-methoxybenzyl methyl ether (**17**) yielded benzoic acid and not the corresponding ester.

The optimised conditions were subsequently used at a preparative scale, with 1 mmol of the selected substrates (see Fig. 3) to confirm the practicality and effectiveness of our aerobic photocatalytic oxidation approach employing Sc(OTf)₃ (see Supplementary Section S3 for details). The reaction progression was monitored using ¹H NMR analysis and the time was optimised to achieve the maximal

conversion of reactants to products. Interestingly, *p*-xylene (**1f**), a substrate with two potential oxidation sites, was oxidised chemoselectively to **2f** at the preparative scale after appropriately adjusting the reaction time.

## Sc(OTf)$_3$-based photocatalytic oxidative cyanation of arenes

A further aim of this study was to demonstrate the utility of Sc-based photoredox catalysis. We chose arenecarbonitriles, which serve as crucial structural scaffolds in synthesising bioactive compounds, alongside their substantial role as intermediates in organic synthesis[41]. We particularly focused on the direct photooxidative cyanation of the aromatic C-H bond, as this reaction remains a formidable challenge for the chemical community because of the need for prefunctionalised starting material and the lack of regioselectivity. Moreover, photocatalytic methodologies for direct C-H cyanation are limited and still require further exploration[42–48].

We first used diphenyl ether (**18a**) ($E_{ox}$ = 1.88 V, vs. SCE)[49] as a benchmark substrate and performed the cyanation using TMSCN as a nucleophilic cyanating agent (12 equiv.) in oxygen–saturated MeCN under blue light irradiation (400 nm) for 24 h. Our preliminary investigations (Fig. 4a, entry 1) showed that a mixture of 4-phenoxybenzonitrile (**19a**) and 2-phenoxybenzonitrile (**19b**) could be obtained in a good yield of 74%, with notable regioselectivity towards the para-substituted isomer. Control experiments demonstrated that the cyanation of **18a** ceased in the absence of light or Sc(OTf)$_3$, or under anaerobic conditions (entries 2-5). Raising the wavelength of the irradiation source from 400 nm to 450 nm resulted in only trace yields of the desired product (entry 6). For explanation, the cyanating agent TMSCN was used in a significant excess (12 equiv.) to avoid competitive photooxidation reactions (see Supplementary Section S4).

Similarly, as was observed for the benzylic oxidations, switching Sc(OTf)$_3$ to Mg(OTf)$_2$, Zn(OTf)$_2$, La(OTf)$_3$, or Ba(OTf)$_2$ resulted in the cessation of cyanation under otherwise identical experimental conditions (see Supplementary Section S4). Switching from MeCN to MeOH or DMSO also completely inhibited the reaction, whereas the reaction in MeNO$_2$ occurred with remarkable conversion. Reaction in dioxane gave small amount of product (Fig. 4a, entries 7-10). Reducing the amount of the Sc(OTf)$_3$ catalyst (2.5 mol%) lowered the yield, whereas doubling the amount (10 mol%) gave a slightly higher yield (entries 11 and 12).

Using the optimised conditions, we next investigated the direct cyanation of various arenes, including both electron-rich and electron-poor substrates, as depicted in Fig. 4b (see Supplementary Section S6 for details). Cyanation of 1,3,5-trimethoxybenzene (**18b**) and 1,3-dimethoxybenzene (**18c**) furnished the corresponding carbonitriles **19c** and **19d**, respectively, as sole products in quantitative yields. Cyanation of 3-chloroanisole (**18d**) resulted in a mixture of 4-chloro-2-methoxybenzonitrile (**19e**) and 2-chloro-4-methoxybenzonitrile (**19f**) in excellent yields, with 3.9:1 regioselectivity. Efficient cyanation of biphenyl (**18e**) occurred, yielding 4-phenylbenzonitrile (**19g**). Cyanation of butoxybenzene (**18f**) led to the formation of 4-butoxybenzonitrile (**19h**) in a quantitative yield; however, the efficiency of this system in the cyanation of electron-poor substrates was only modest. Cyanation of methyl 2-methoxybenzoate (**18g**) and 1-methoxy-2-(trifluoromethyl) benzene (**18h**) gave poor yields of the desired products **19i** and **19j**, respectively. The highly electron-poor substrates, such as chlorobenzene (**18i**), methyl benzoate (**18j**), and trifluorotoluene (**18k**), did not yield any cyanation products. Subsequent application of the optimised conditions using 1 mmol of the selected substrates **18a** and **18d** led to cyanation products in good yields, thereby confirming the practicality of our cyanation approach on a preparative scale.

## Mechanistic investigations

Additional control experiments (see Figs. 2a and 4a for the first set of control experiments) were first conducted to unravel the mechanisms

underlying Sc(OTf)$_3$-based photooxidative transformations. Toluene oxidation was clearly impeded in the presence of TEMPO (2,2,6,6-tetramethylpiperidine-1-oxyl) as a radical scavenger. Atmospheric pressure chemical ionization mass spectrometry (APCI-MS) data confirmed the formation of a TEMPO-benzyl adduct (Fig. 5a), indicating the involvement of a radical pathway in this oxidation process. Similarly, the cyanation of diphenyl ether was almost completely halted in the presence of TEMPO (see Supplementary Section S7 for details).

Oxidation of toluene using isotopically labelled oxygen $^{18}O_2$ gave mainly benzoic acid (**2a**) with two incorporated $^{18}O$ atoms and almost no product containing two $^{16}O$ atoms (Fig. 5b). This result supports the hypothesis that molecular oxygen is the source of the oxygen atom in the product. Some amount (30 %) of product containing one unlabelled $^{16}O$ atom and one $^{18}O$ was also observed which is explained by enhanced residual moisture causing oxygen atom exchange in benzaldehyde intermediate by oxo –hydrate form equilibria which is supported by Sc$^{3+}$ (Lewis acid) catalysis (see Supplementary Section S8 for details).

Stern-Volmer quenching experiments revealed that toluene, diphenyl ether and 1,3,5-trimethoxybenzene quenched the fluorescence of a Sc(OTf)$_3$ solution in acetonitrile with Stern-Volmer constants $K_s$ of 3, 22 and 42 L mol$^{-1}$, respectively (see example in Fig. 5c). This signified SET from the tested substrates to an excited scandium complex (see Supplementary Section S7 for details).

As mentioned earlier, the UV-Vis absorption spectrum of Sc(OTf)$_3$ in acetonitrile showed an elongated tail in the visible region around 400 nm and a fluorescence emission spectrum with a sharp band at 505 nm (Fig. 2d). ESI-MS measurements aimed at elucidating the structure of the scandium complex responsible for this absorption in a solution of Sc(OTf)$_3$ in MeCN. It suggested the formation of a cluster of complexes, including [Sc(OTf)$_2$(MeCN)$_2$]$^+$, as the primary species. However, these particles do not absorb in the visible region but below 260 nm, as indicated by the theoretical spectra (see Supplementary Section S12 and Supplementary Data 1 for Cartesian coordinates). By contrast, particles containing coordinated molecular oxygen seems to absorb visible light. For example the [Sc(MeCN)$_3$(($\eta^2$-O$_2$)]$^{3+}$ particle in S$_0$ state, a complex with three MeCN molecules and coordinated oxygen molecule could absorb in the visible region (400 – 442 nm) as documented by the theoretical spectrum, consistent with our experimental results (Fig. 5d). Moreover, calculated emission of this species (522 nm) corresponds to experimental fluorescence (around 506 nm; see Supplementary Section S12).

Indeed, the coordination of oxygen molecule is ascribed to the oxophilicity of Sc$^{3+}$ ions and is further supported by our EPR and cyclic voltammetry measurements (see below). It should be noted that the isomeric [Sc(MeCN)$_3$($\eta^1$-O$_2$)]$^{3+}$ complex (in T$_1$ state) is more stable by 0.950 eV (91.67 kJ mol$^{-1}$) at DLNPO-CCSD(T)/aug-cc-pVTZ level of theory (see Supplementary Section S12 for further information on excited state energies) compared to the S$_0$ state [Sc(MeCN)$_3$(($\eta^2$-O$_2$)]$^{3+}$, nevertheless, [Sc(MeCN)$_3$($\eta^1$-O$_2$)]$^{3+}$ is proposed to not absorb in the visible region (absorption below 300 nm, see Supplementary Section S12). We believe the small barrier can be overcome thermally.

Interaction of Sc$^{3+}$ ions with molecular O$_2$ was also supported by cyclic voltammetry experiments (Fig. 5e). Addition of Sc(OTf)$_3$ into oxygen-saturated acetonitrile was characterised by formation of a new peak ($E_{cp}$ = −0.95 V vs SCE) with less negative potential compared to the peak of oxygen in acetonitrile ($E_{pc}$ = −1.16 V vs SCE). The new signal is attributed to oxygen coordinated to Sc$^{3+}$ ions. The same signal can be achieved by an inverse experiment. Cyclic voltammogram of Sc(OTf)$_3$ under argon contains peak at −1.83 V vs SCE corresponding to Sc$^{3+}$ reduction. After bubbling the solution with oxygen gas, two peaks attributed to coordinated and "free" oxygen appeared (see Supplementary Section S9).

In subsequent experiments, we investigated the mechanism of Sc-based photooxidative catalysis using toluene oxidation as the

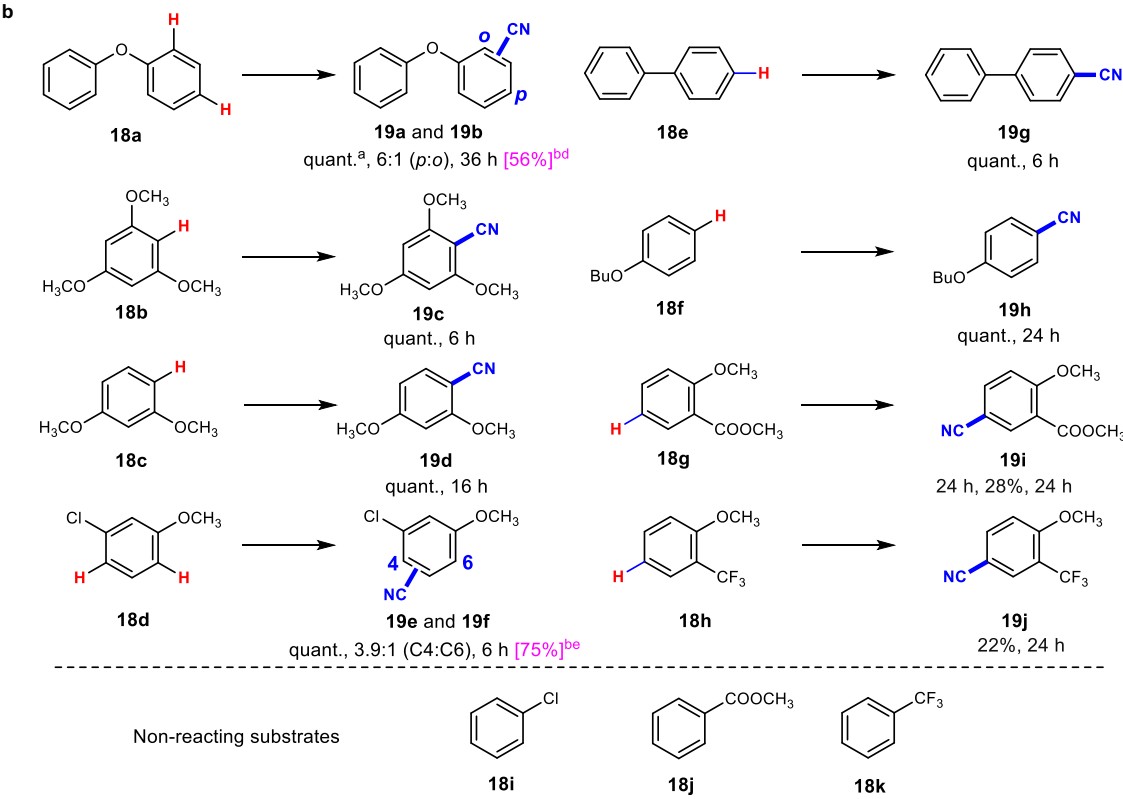

**a**

| Entry | Deviations from Optimized Conditions | Yield[b] [%] | |
| --- | --- | --- | --- |
| | | 4-CN [%] | 2-CN [%] |
| 1 | None | 74 | traces |
| 2 | No Sc(OTf)₃ | n.d. [c] | n.d. [c] |
| 3 | No light | n.d. [c] | n.d. [c] |
| 4 | Air instead of O₂ | n.d. [c] | n.d. [c] |
| 5 | Argon instead of O₂ | n.d. [c] | n.d. [c] |
| 6 | 450 nm instead of 400 nm | traces | n.d. [c] |
| 7 | MeOH instead of MeCN | n.d. [c] | n.d. [c] |
| 8 | DMSO instead of MeCN | n.d. [c] | n.d. [c] |
| 9 | MeNO₂ instead of MeCN | 68 | traces |
| 10 | 1,4-dioxane instead of MeCN | 10 | n.d. [c] |
| 11 | 2.5 mol% instead of 5 mol% | 46 | n.d. [c] |
| 12 | 10 mol% instead of 5 mol% | 83 | n.d. [c] |

**Fig. 4 | Experiments investigating Sc(OTf)₃-based photocatalytic oxidative cyanation of arenes under aerobic conditions. a** Condition optimisation and control experiments. **b** Reaction scope for arenes. [a] Standard reaction conditions: Substrate (0.140 mmol), Sc(OTf)₃ (5 mol%), Me₃SiCN (12 equiv., 1.68 mmol), MeCN (0.25 mL), 45 °C, O₂ (balloon), blue LED (400 nm), and 24 h (unless otherwise specified); Yields were determined by ¹H NMR unless otherwise stated. [b] Isolated yield (in magenta); experiments were performed at a 1 mmol scale (see Supplementary Section S6 for details). [c] Product was not detected. [d] 48% of **19a**, and 8% of **19b**. [e] total yield of a mixture of **19e** and **19 f** (4:1).

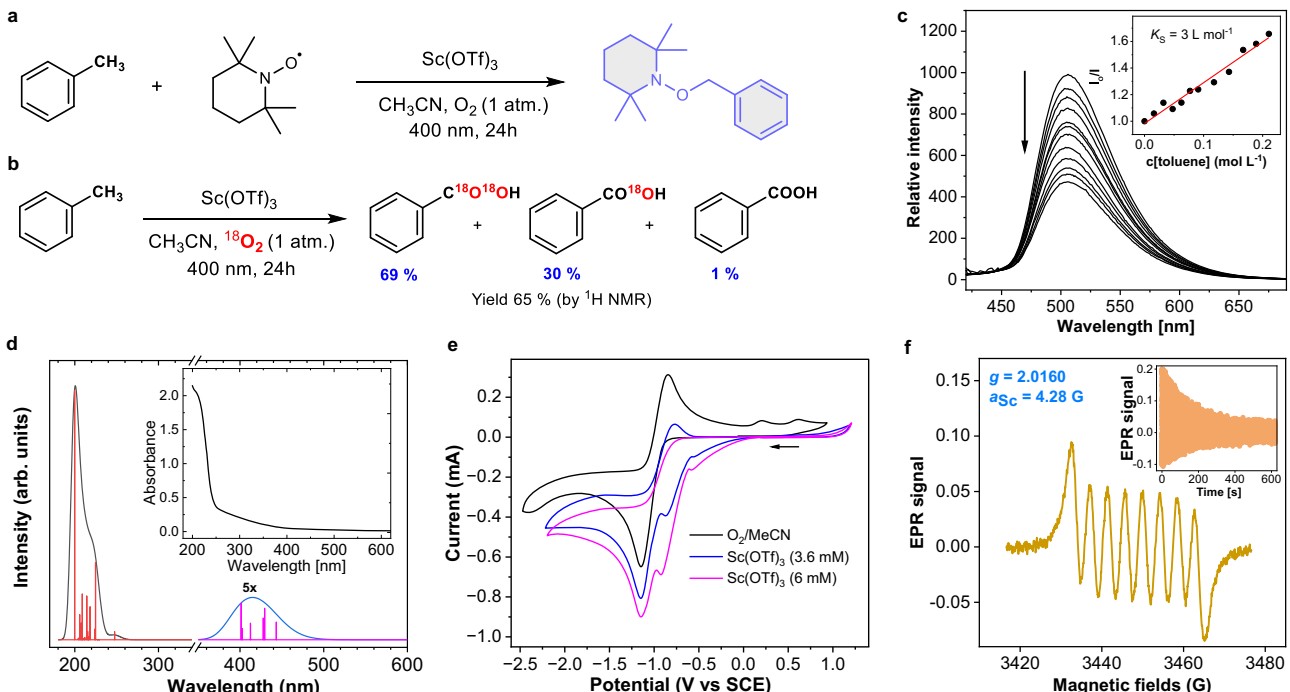

**Fig. 5 | Mechanistic studies. a** Formation of a TEMPO-benzyl adduct during photooxidation of toluene using Sc(OTf)$_3$. **b** Oxidation with isotopically labelled oxygen (see Supplementary Section S8 for details). **c** Quenching of fluorescence of Sc(OTf)$_3$ solution in MeCN ($c = 0.05$ M, $\lambda_{ex} = 400$ nm) with toluene and the corresponding Stern-Volmer plot (inset). **d** Comparison of the theoretical absorption spectra of [Sc(CH$_3$CN)$_3$(O$_2$)]$^{3+}$ obtained by quantum chemical calculations at PBE0/Def2-TZVPD level of theory (DFT calculations were performed in Orca software package, release 6.0.1., ref. 56) with experimental spectra of Sc(OTf)$_3$ solution in MeCN (inset). **e** Cyclic voltammogram at Glassy Carbon microdisc working electrode of oxygen-saturated acetonitrile before and after addition of Sc(OTf)$_3$. **f** The EPR spectrum of an oxygen-saturated MeCN solution of toluene ($c = 2$ M) and Sc(OTf)$_3$ ($c = 0.1$ M) upon irradiation with a 400 nm LED at room temperature; kinetic trace after stopping the irradiation (inset).

representative example. Irradiation of an oxygen-saturated acetonitrile solution containing Sc(OTf)$_3$ (0.1 M) and toluene (2 M) with a 400 nm LED at room temperature resulted in a distinct isotropic EPR signal (Fig. 5f). The pronounced eight-line isotropic spectrum was attributed to the formation of a complex of superoxide radical anion with scandium ion, Sc$^{3+}$-O$_2$$^{•-}$. This resulted in superhyperfine splitting ($a_{Sc} = 4.28$ G) due to the 7/2 scandium nuclear spin and featured an isotropic g value, $g_{iso} = 2.0160$. This observation is consistent with reported EPR spectra in the literature[30].

Interestingly, while previous studies required a low temperature to observe Sc$^{3+}$-O$_2$$^{•-}$ by EPR, we were able to obtain well-defined spectra at room temperature. The kinetic recordings showed stability of the Sc$^{3+}$-O$_2$$^{•-}$ complex for about 2 min after the cessation of irradiation (Fig. 5f, insight). Importantly, no significant EPR signal appeared upon irradiation of a blank solution containing no toluene. Irradiation of the acetonitrile solution of Sc(OTf)$_3$ and toluene under anaerobic conditions also did not reveal any EPR signals. Replacing Sc(OTf)$_3$ with La(OTf)$_3$ (which showed no catalytic activity in photooxidative procedures) led to a very faint EPR signal, even after an extended period of irradiation (see Supplementary Section S10). These observations provide evidence that all species − toluene as a substrate and electron donor, oxygen as a terminal electron acceptor and scandium as mediator − are necessary for efficient photoinduced electron transfer and Sc$^{3+}$-O$_2$$^{•-}$ formation.

### Transient absorption spectroscopy

We utilised ultrafast transient absorption (TA) spectroscopy to unravel the nature of the excited state species involved. We performed measurements of Sc(OTf)$_3$ in MeCN under both O$_2$ and N$_2$ atmospheres (purging for 30 minutes, followed by gassing with N$_2$ in a purity of 99.99%). Initially, measuring a solution of Sc(OTf)$_3$ in MeCN under an

inert N$_2$ atmosphere resulted in a strong bleach signal at approximately 500 nm, accompanied by weak excited-state absorption (ESA) features (Fig. 6a). This bleach rapidly converted into a positive signal over a slower time scale of a few nanoseconds. The positive signal then continuously grew over the microsecond time scale until reaching a maximum at around 1 microsecond, after which it faded slowly, as depicted in Fig. 6b-c. In the presence of O$_2$, similar kinetic behaviour was found at the early stage, but with a feeble TA signal, and only during the long timescale (microseconds, the decay was somewhat faster than this in the N$_2$ atmosphere) (see Fig. 6d). The initial bleach signal at 500 nm was assigned to the initial fluorescence of the Franck-Condon (FC) state, which quickly relaxed to some non-emissive state (S$_1$ relaxed); this took 300 fs. This relaxed state populated another state, the triplet state, at a slow rate (10 ns). The triplet state was then slowly quenched over a microsecond lifetime.

The weak invariance between the N$_2$ and O$_2$ atmosphere was attributed to the O$_2$ that was strongly bound to the Sc$^{3+}$ complex and present even under the N$_2$ atmosphere because of strong Sc oxophilicity[35]. A role for the triplet state of O$_2$ in the population of the Sc$^{3+}$ complex triplet state was highly expected.

When measured in toluene/MeCN (Fig. 6e-h), the transient absorption data were changed to a greater extent than those in pure MeCN, but remained similar under O$_2$ and N$_2$ atmospheres. For instance, the initial fluorescence from FC was missing, but a fast lifetime decay of 500 fs (due to the relaxation state) was present, followed by a lifetime rise of 20 ps that was assigned to the early formation of the triplet state (see Fig. 6e). Within 20 ns, the bleach signal around ca. 500 nm started to evolve again, followed by an increase in the overall ESA features across the visible region over 90 ns. The appearance of the bleach signal was assigned to the delayed fluorescence of the Sc complex, which can occur due to the reversible ISC (r-ISC) process

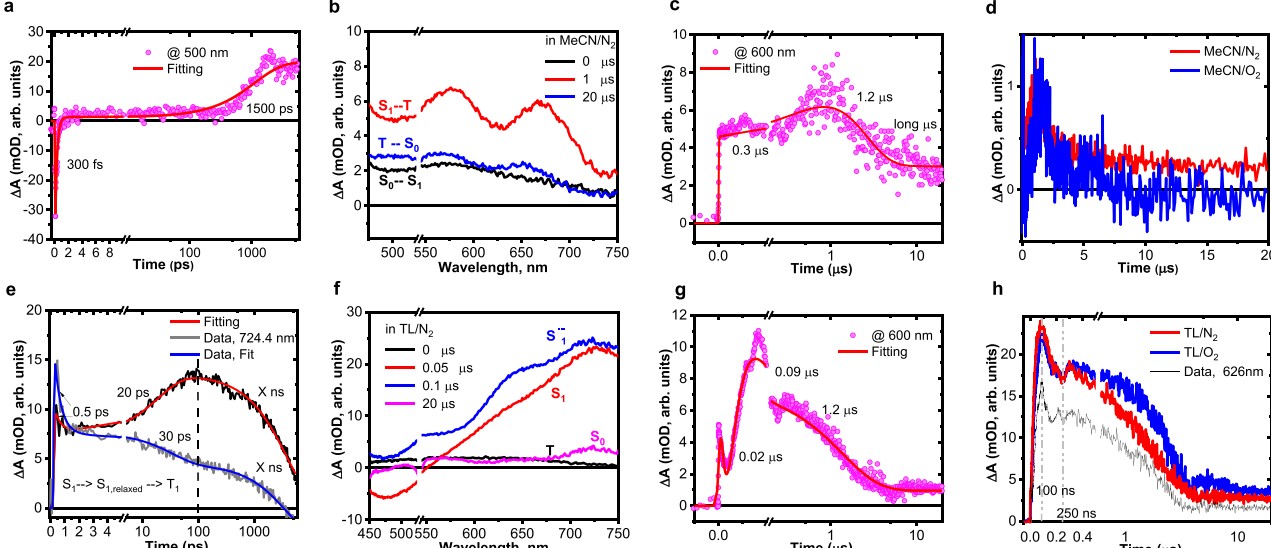

**Fig. 6 | Mechanistic studies by transient absorption spectroscopy. a** Extracted kinetic trace @ 500 nm for the excited $Sc(OTf)_3$ in MeCN under $N_2$ (from fs to ns). **b** Extracted spectra for the excited $Sc(OTf)_3$ in MeCN under $N_2$ at longer time scale (μs), each spectrum is assigned to the population of an excited state. **c** Extracted kinetic trace from $Sc(OTf)_3$ in MeCN under $N_2$ at longer time scale (μs). **d** Comparison between extracted kinetic traces under $N_2$ and $O_2$ for $Sc(OTf)_3$ in MeCN at longer time scale. **e** Extracted kinetic trace for $Sc(OTf)_3$ in toluene under $O_2$ at short time scale (fs-ns). **f** Evolving spectra for $Sc(OTf)_3$ in toluene under $O_2$ at longer time scale (μs). **g** Extracted kinetic trace for $Sc(OTf)_3$ in toluene under $O_2$ at longer time scale (μs). **h** Comparison between extracted kinetic traces under $N_2$ and $O_2$ for $Sc(OTf)_3$ in toluene.

from the triplet to the singlet state[50], as depicted in Fig. 6f. Due to the presence of toluene molecules, an ET was expected to occur from toluene to the Sc complex on a time scale of 90 ns, leading to the disappearance of the fluorescence peak and formation of $S_1^-$ state, as evident from Fig. 6f-g. The recombination of this reduced state to the $S_0$ state occurred at a slow rate > 1.2 microseconds.

The absence of toluene remarkably affected the Sc complex behaviour by delaying the triplet state formation in MeCN vs in the MeCN/toluene mixture. The insensitivity of the Sc complex towards the $O_2$ atmosphere also highlighted the role of strongly bound $O_2$ molecules in extending the lifetime of the observed excited states. For instance, the slow decay of the $S_1^-$ state was expected to be due to structural rearrangements within the Sc complex, leading to the desorption of the reduced $O_2$ into the reaction mixture for further use in product formation pathways.

## Mechanism of $Sc(OTf)_3$-based photocatalysis

Drawing from the outcomes of control experiments, mechanistic studies using EPR and transient absorption measurements (TA), and considering previous reports, a plausible mechanism for the $Sc(OTf)_3$-catalysed photooxidation reactions is postulated (Fig. 7). Initially, the $Sc^{3+}$ complex can be excited upon coordination with molecular oxygen, as demonstrated by the quantum chemical calculations (see Fig. 5d and Supplementary Section S12). This type of complex is proposed to involve acetonitrile ligands and the participation of triflates either as counterions or ligands. Upon excitation, the scandium complex forms a singlet excited state, which can subsequently undergo an intersystem crossing (ISC) process, as elucidated by TA, to generate a triplet excited state of the scandium complex. Nevertheless, according to TA measurements, in the presence of toluene, a delayed-singlet excited state of the $Sc^{3+}$ complex (with oxygen involved) seems to be responsible for the productive reaction pathway that undergoes SET from toluene during the production of a toluene radical cation and the $Sc^{3+} \cdot O_2^{-}$ complex, as verified by the eight-line EPR spectrum arising from the I = 7/2 $^{45}$Sc nucleus. Subsequently, the superoxide ion, $O_2^{-}$ abstracts a proton (proton transfer = PT) from the benzyl radical cation to produce a hydroperoxy radical, $HOO^{·}$ and a benzyl radical.

Subsequently, the interaction between the benzyl radical and $O_2$ produces the peroxyl radical, followed by benzaldehyde formation. Further oxidation of benzaldehyde yields the desired product, benzoic acid (**2a**), as well as $H_2O_2$, as confirmed by iodometry (see Supporting Section S11 for details). Alternatively, the aryl radical cation can react with a cyanide source (TMSCN) to generate a cyclopentadienyl radical and produce a cyanoarene by hydrogen atom transfer (HAT). The final transformation using the hydroperoxy radical is likely the one proposed by Nicewicz[48] but an alternative participation of oxygen is not excluded.

From mechanistic point of view, our method using redox-innocent $Sc^{3+}$ cation represents another mode of metal ion functioning in oxidative catalysis and brings an alternative to (i) Ir or Ru complexes with high oxidation power[8–10], and (ii) to the recently developed systems based on ligand-to-metal charge transfer employing halides of metals like Ce, Fe, or Bi. The latter method uses halogen or alkoxy radical generation which causes hydrogen atom transfer[51–55]. Method with $Sc^{3+}$ is based on MCET and as such, it has some limitations given by electron transfer thermodynamics. Nevertheless, on the example of 4-(trifluoromethyl)toluene (**1c**) oxidation, we have demonstrated possible overcoming this limit by the presence of a proton source (trifluoroacetic acid).

## Discussion

In conclusion, all the results obtained during this study confirm that $Sc(OTf)_3$ possesses noteworthy photocatalytic activity. Hence, we introduce an aerobic oxidation approach for benzylic substrates with $Sc(OTf)_3$ as the sole photocatalyst. This system revealed considerable efficiency with a wide range of benzylic substrates, including alcohols, toluene derivatives, and methylene-containing substrates. A straightforward $Sc(OTf)_3$-based photocatalytic approach for the oxidative cyanation of arenes under aerobic conditions was also developed as an example of direct aromatic C-H derivatization resulting in C-C bond formation. The photooxidative cyanation method, despite some limitations (e.g., the cyanation of electron-poor substrates), demonstrated an interesting efficiency with a notable regioselectivity across the tested scope of arenes.

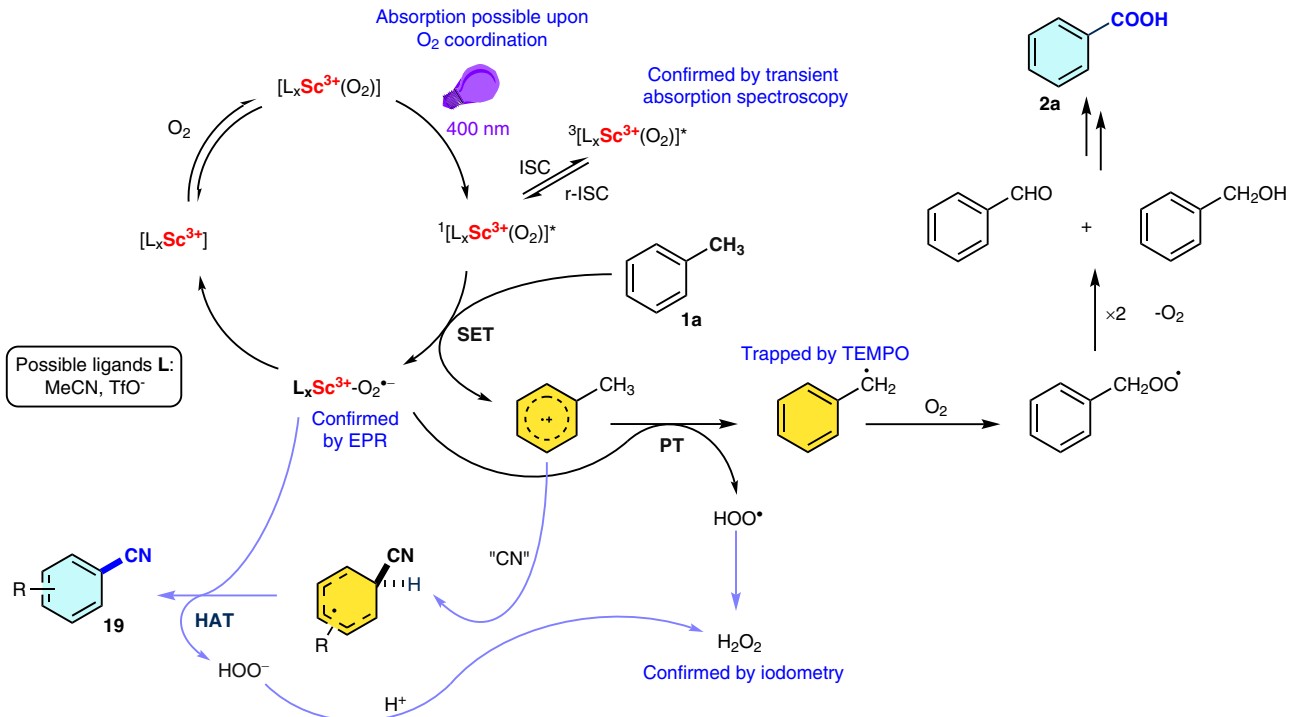

**Fig. 7 | Proposed mechanism of aerobic oxidative transformations catalysed by Sc³⁺ complexes in acetonitrile.** The mechanism involves excitation of Sc³⁺ species upon oxygen coordination, intersystem crossing (ISC), singlet electron transfer (SET) from a substrate (toluene, **1a**) and transformation of thus formed radical cation either by (i) by proton transfer (PT) followed by the reaction with oxygen to form oxygenation product **2a**, or (ii) by the reaction with cyanide source followed by hydrogen atom transfer (HAT) to form benzonitrile **19**.

Our results on benzylic photooxidation and oxidative cyanation mediated by Sc(OTf)₃ as sole photocatalyst break the long-held belief that Sc³⁺ salts serves only as a potent Lewis acids for activating organophotocatalysts or substrates. We expect that Sc³⁺ salts can find applications as photocatalysts also in other oxidative organic reactions, some of which are now under investigation in our laboratory. From a general perspective, our examples could pave the way for the exploration of other redox-innocent metals in photoredox catalysis.

## Methods

### General procedure for photooxidation of benzylic substrates
**Small scale experiments (A).** A vial was charged with a mixture of the substrate (140 μmol) and Sc(OTf)₃ (5 mol%, 7 μmol) in MeCN (250 μL). The reaction mixture was bubbled with oxygen (2 min) and then stirred at 45 °C under irradiation with 400 nm LEDs. After selected time, the reaction mixture was diluted with DMSO-d₆ and the yield was determined by ¹H NMR.

**Preparative experiments (B).** A mixture of the substrate (1 mmol) and Sc(OTf)₃ (5 mol%) in MeCN (5 mL) was bubbled with oxygen (2 min). It was then stirred at ambient temperature in a 50 mL Schlenk tube under irradiation with 400 nm LEDs under aerobic conditions (oxygen balloon). The reaction progress was monitored by ¹H NMR. When the reaction was completed, the solvent was evaporated, and the crude product was purified either by column chromatography on silica gel or by extraction (see Supplementary Section S3 for details).

### General procedure for photocatalytic cyanation of arenes
**Small scale experiments (C).** A vial was charged with a mixture of the substrate (140 μmol), Sc(OTf)₃ (5 mol%, 7 μmol) and TMSCN (12 equiv., 1.68 mmol) in MeCN (250 μL). The reaction mixture was bubbled with oxygen (2 min) and then stirred at 45 °C under irradiation with 400 nm LEDs. After selected time, the reaction mixture was diluted with DMSO-d₆ and the yield was determined by ¹H NMR.

**Preparative experiments (D).** A mixture of the substrate (1 mmol), Sc(OTf)₃ (5 mol%) and TMSCN (12 equiv., 12 mmol, 1501 μL) in MeCN (5 mL) was bubbled with oxygen (2 min). It was then stirred at ambient temperature in a 50 mL Schlenk tube under irradiation by 400 nm Luxeon LEDs under aerobic conditions (oxygen balloon). The reaction progress was monitored by TLC and ¹H NMR. When the reaction was completed, the solvent was evaporated, and the crude product was purified by column chromatography on silica gel (see Supplementary Section S6 for details).

## Data availability
The data supporting the results of the article, including optimization studies, experimental procedures, compound characterization, mechanistic studies, and details on quantum chemical calculations are provided within Supplementary Information and Supplementary Dataset. All information (raw, processed and visualized data) about prepared compounds, techniques used for characterisation of prepared compound, NMR, MS, spectral and other used methods are available in this dataset: https://doi.org/10.5281/zenodo.14328398. Data supporting the findings of this manuscript are also available from the corresponding author upon request.

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

## Acknowledgements

This work was supported by the project "The Energy Conversion and Storage", funded as project No. CZ.02.01.01/00/22_008/0004617 by Programme Johannes Amos Comenius, call Excellent Research. Computational resources were provided by the e-INFRA CZ project (ID:90254), supported by the Ministry of Education, Youth and Sports of the Czech Republic.

## Author contributions

A.H.T. and R.C. conceived the idea. A.H.T. was responsible for designing and conducting experimental procedures and measurements, analysing the data, and interpreting the results. A.E.-Z. and J.I.K. performed transient absorption experiments. A.E.-Z. contributed to discussion of mechanism. J.S. performed the DFT calculations. J.K. and K.L. contributed to EPR measurements and their evaluation. J.C. performed MS measurements. M.P. performed cyclic voltammetry measurements. E.S. provided data managements and investigation and the corrections of the supplementary information and provided measurements with $^{18}O_2$. A.H.T., A.E.-Z., and R.C. collaborated in drafting the manuscript. R.C. directed the project.

## Competing interests

The authors declare no competing interests.
