## [Transparent Peer Review file · Nature Communications]

Redox-innocent scandium(III) as the sole catalyst in visible light photooxidations

Corresponding Author: Professor Radek Cibulka

Version 0:

Reviewer comments:

Reviewer #1

(Remarks to the Author)

The authors describe the use of scandium(III) triflate, $\text{Sc}(\text{OTf})_3$, as a sole photocatalyst in visible light-driven reactions. The two reactions developed in this manuscript, which are not the most compelling reactions for showcasing a new concept, are the direct aerobic oxidative C-H functionalization of aromatic substrates and the direct cyanation of aromatic rings. Nevertheless, the concept of using $\text{Sc}(\text{OTf})_3$ challenges the dogma that it is limited to its traditional role as a Lewis acid.

The biggest strength of the manuscript may be the detailed mechanistic studies, including EPR and transient absorption spectroscopy. Based on these experiments, the authors propose a radical pathway involving a scandium-oxo complex. The study also suggests the potential for other $\text{Sc}(\text{OTf})_3$ -catalyzed photooxidation reactions. The findings broaden the potential applications of scandium salts in photocatalytic processes, and it opens new avenues for exploring other "redox innocent" metals in photoredox catalysis.

Overall, this is a well-written manuscript. References are complete. The supporting information is thorough and supports the conclusions of the manuscript. Given the conceptual novelty of developing scandium-based photocatalysts, the research presented in this manuscript will be of great interest to the readers of Nature Communications. I recommend that this manuscript be published as is.

Reviewer #2

(Remarks to the Author)

In this paper, Cibulka and co-workers report a scandium triflate-catalyzed, visible light-induced aerobic oxidative C-H functionalization of aromatic substrates. This work is noteworthy for utilizing scandium triflate as a photocatalyst, a compound typically employed as a Lewis acid catalyst. A catalytic cycle is proposed, supported by several control experiments and a Stern-Volmer fluorescence quenching experiment. Overall, the results are clearly of interest, and the experimental data are well-organized. However, in the opinion of this reviewer, the study unfortunately lacks the groundbreaking character required for publication in Nature Communications. The reasons for this are outlined below:

- Oxidative C-H functionalization of aromatic substrates has already been achieved with FeCl_3 and CeCl_3 (Chin. J. Chem. 2021, 39, 3225-3230; Science China Chemistry 2021, 64, 1487-1492). These two catalysts exhibit higher efficiency than scandium triflate. The method presented in this study does not offer a significant improvement over these existing methods.
- While your work appears technically sound, insights from the studies are quite specialized with a narrow scope of the core structure (electron-rich aromatic substrates) and do not present sufficient conceptual advancement for publication at Nature Communications.
- The $[\text{Sc}(\text{MeCN})_3(\eta^2\text{-O}_2)]^{3+}$ species is proposed to be the active catalyst in this transformation. However, the authors do not provide any direct evidence to support this claim.

Reviewer #3

(Remarks to the Author)

This study presents the first report on the catalytic activity of $\text{Sc}(\text{OTf})_3$ as a standalone photoredox catalyst, challenging the conventional perception that Sc^{3+} solely functions as a Lewis acid. This breakthrough pioneers a new direction for the

application of rare-earth metals in photocatalysis. The work demonstrates the high efficiency and broad substrate compatibility of $\text{Sc}(\text{OTf})_3$ in benzylic oxidation and direct aromatic ring cyanation, showcasing its promising practical potential. Mechanistic insights were rigorously validated through multiple approaches, including EPR spectroscopy, transient absorption spectroscopy, and theoretical calculations, significantly reinforcing the credibility of the conclusions. The reaction was sophisticatedly designed and well conducted. The results reported in this paper are both significant and intriguing. This reviewer would like to recommend acceptance of this work after the following issues are addressed.

1. The applicable scope of this reaction is limited, especially for the cyanation of arenes. In addition, regarding the low yield of substrates such as 4-(trifluoromethyl)toluene (1c), it is necessary to further discuss the relationship between its electronic structure and oxidation potential, or attempt to introduce co-catalysts to improve the performance.
2. The author mentioned that O_2 acts as both an electron acceptor and a reactant. It is proposed to further verify the specific participation mode of oxygen through isotope-labeling experiments (such as using $^{18}\text{O}_2$).
3. Supplement with the latest progress on "metal ion-coupled electron transfer (MCET)" in recent years (such as literatures after 2020) to reflect the timeliness of the research.
4. Add DFT calculations of the excited-state electronic structure to clarify the correlation between light absorption and catalytic activity.

Version 1:

Reviewer comments:

Reviewer #3

(Remarks to the Author)

After careful review of the revised manuscript and the authors' detailed responses to all prior comments, I am satisfied that the concerns raised have been adequately addressed. The authors have demonstrated a thorough understanding of the suggestions and have implemented the modifications thoughtfully, which significantly strengthens the validity and clarity of the study. The revised content now meets the standards of the journal, and I therefore recommend this manuscript for acceptance.

The authors are grateful for all comments to the manuscript. Herein we address the issues raised by the reviewers and editorial office. Our response to the reviewers is highlighted in blue. Corresponding revisions carried out in the manuscripts are highlighted with a yellow background.

Reviewer 1:

The authors describe the use of scandium(III) triflate, Sc(OTf)₃, as a sole photocatalyst in visible light-driven reactions. The two reactions developed in this manuscript, which are not the most compelling reactions for showcasing a new concept, are the direct aerobic oxidative C-H functionalization of aromatic substrates and the direct cyanation of aromatic rings. Nevertheless, the concept of using Sc(OTf)₃ challenges the dogma that it is limited to its traditional role as a Lewis acid.

The biggest strength of the manuscript may be the detailed mechanistic studies, including EPR and transient absorption spectroscopy. Based on these experiments, the authors propose a radical pathway involving a scandium-oxo complex. The study also suggests the potential for other Sc(OTf)₃-catalyzed photooxidation reactions. The findings broaden the potential applications of scandium salts in photocatalytic processes, and it opens new avenues for exploring other "redox innocent" metals in photoredox catalysis.

Overall, this is a well-written manuscript. References are complete. The supporting information is thorough and supports the conclusions of the manuscript. Given the conceptual novelty of developing scandium-based photocatalysts, the research presented in this manuscript will be of great interest to the readers of Nature Communications. I recommend that this manuscript be published as is.

The authors are very grateful for the positive evaluation. We are especially pleased that the reviewer highlights the essence of the manuscript, which is aimed at emphasizing the novel role of Sc(OTf)₃ in photoredox catalysis as a photoactive catalyst itself and at refuting the "dogma" that the use of Sc(OTf)₃ is limited to its traditional role as a Lewis acid.

Reviewer 2:

In this paper, Cibulka and co-workers report a scandium triflate-catalyzed, visible light-induced aerobic oxidative C-H functionalization of aromatic substrates. This work is noteworthy for utilizing scandium triflate as a photocatalyst, a compound typically employed as a Lewis acid catalyst. A catalytic cycle is proposed, supported by several control experiments and a Stern-Volmer fluorescence quenching experiment. Overall, the results are clearly of interest, and the experimental data are well-organized. However, in the opinion of this reviewer, the study unfortunately lacks the groundbreaking character required for publication in Nature Communications. The reasons for this are outlined below:

The authors thank for evaluation by reviewer 2. Like the other two reviewers, we believe that the submitted manuscript provides insights of fundamental importance and are therefore suitable for the journal Nature Communications. We would like to note that the aim of the manuscript was not to develop the most efficient system for the selected model reactions, but to demonstrate a completely new and unexpected role of Sc(OTf)₃. This salt has so far been considered and used exclusively as a Lewis acid. Even in photoredox catalysis, as evidenced by a number of citations in the original manuscript (citations 12-15) and also by recent review articles, which we have added to the revised version [ref 16 = *Org Chem Front* (2025), DOI 10.1039/D5QO00102A; ref. 26 = *JACS Au* **4**, 344-368

(2024); ref. 17 = *Bull. Korean Chem. Soc.* **45**, 503-519 (2024); ref. 18 = *Asian J. Org. Chem.* **13**, e202400295 (2024)]. In our manuscript, we refute this dogma and show that in photoredox catalysis, Sc(OTf)₃ can act as a completely independent catalyst excitable by visible light. We expect that the new perspective of the chemical community on Sc(III) salts will allow a significant expansion of their use in further transformations and will also inspire them to research similar redox-innocent metal ions. We are aware that in the original manuscript we may not have emphasized the main goal of our article clearly enough and therefore we have added the following text to the revised manuscript: "In this work we demonstrate the previously unknown activity of Sc(OTf)₃ functioning as a sole photoredox catalyst."

Statement on specific issues:

a) *Oxidative C-H functionalization of aromatic substrates has already been achieved with FeCl₃ and CeCl₃ (Chin. J. Chem. 2021, 39, 3225-3230; Science China Chemistry 2021, 64, 1487-1492). These two catalysts exhibit higher efficiency than scandium triflate. The method presented in this study does not offer a significant improvement over these existing methods.*

The authors thank the reviewer for reminding us of the oxidations using CeCl₃ and FeCl₃. We apologize that we did not cite these articles in our original manuscript. However, it is necessary to take into account that the systems with halides of metal ions like Ce, Fe and others, although often very effective, works by a different mechanism, using LMCT producing a chloride radical or radical of halogenated alcohol, which subsequently generates the corresponding radicals by hydrogen abstraction [*Chem. Commun.* **61**, 1944-1961 (2025)]. This concept is relatively new and certainly deserves to be discussed in our article, because it includes metal ions. We have included this discussion (see below) together with the references recommended by reviewer 2 [*Chin. J. Chem.* **39**, 3225-3230 (2021); *Science China Chemistry* **64**, 1487-1492 (2021)], a further reference to a very recent comprehensive review [*Chem. Commun.* **61**, 1944-1961 (2025)] and a reference to the other selected recent articles [*Beilstein J. Org. Chem.* **17**, 1727-1732 (2021); *J. Phys. Chem. Lett.* **15**, 6202-6208 (2024)] into mechanistic part of our manuscript.

However, as already mentioned above, the goal of our manuscript is not to find the most effective system for benzylic oxidations, but to present a completely new concept using Sc(OTf)₃ as a photocatalyst and, at the same time, to show that Sc(III) ions can function in a role other than Lewis acids. Our system is based on the direct excitation of Sc(III) in the form of a complex formed in the reaction mixture. Great advantage of our catalytic system is that it does not require additional additives which are usually required in the case of systems based on LMCT. *Very importantly, the functionality of the system with Sc(OTf)₃ has been demonstrated also for direct aromatic cyanations which are not achievable by CeCl₃ or FeCl₃, as we proved in a series of independent experiments.*

Text added to mechanistic part of our manuscript: "From mechanistic point of view, our method using redox-innocent Sc³⁺ cation represents another mode of metal ion functioning in oxidative catalysis and brings an alternative to (i) Ir or Ru complexes with high oxidation power,⁷⁻¹⁰ and (ii) to the recently developed systems based on ligand-to-metal charge transfer employing halides of metals like Ce, Fe, or Bi. The latter method uses halogen or alkoxy radical generation which causes hydrogen atom transfer.⁵²⁻⁵⁶ Method with Sc³⁺ is based on MCET and as such, it has some limitations given by electron transfer thermodynamics. Nevertheless, on the example of 4-(trifluoromethyl)toluene (**1c**) oxidation, we have demonstrated possible overcoming this limit by the presence of a proton source (trifluoroacetic acid)."

b) *While your work appears technically sound, insights from the studies are quite specialized with a narrow scope of the core structure (electron-rich aromatic substrates) and do not present sufficient conceptual advancement for publication at Nature communications.*

Regarding this reviewer's remark, we would like to state that due to the pioneering work, we have chosen only model examples to demonstrate a new approach in photoredox oxidative catalysis. Selection of reactions has been motivated by an effort to show two different transformations with completely different products (products of oxidation with C-O bond formation and products of C-C coupling during cyanation). We do not agree with the reviewer that the method is demonstrated on a limited range of substrates. The substrate scope includes oxidations of benzyl alcohols substituted with all types of substituents. In the case of toluenes, we tested electron-rich, electron-neutral and halogen-substituted toluenes considered electron-deficient. On the other hand, the method has its limitation, which is common in PET-based photoredox catalysis. In PET oxidative processes, the transformations are limited by the oxidation potentials of the substrates and by the reduction potentials of the catalysts. This is also the case here. We added the corresponding discussion to the manuscript (also at the request of reviewer 3, see below). In addition, as part of the revision of the article, we found conditions under which the oxidation of electron-deficient substrate like 4-trifluoromethyltoluene ($E_{ox} = +2.61$ V vs SCE) can be initiated. This occurs after the addition of a relatively weak acid (0.2 equiv. only). Most probably, this acid further favours MCET on coordinated oxygen by its subsequent protonation, facilitating the formation of hydrogen peroxide. We also added this result to the manuscript into Figure 3.

Added text: "Substrates with higher oxidation potentials than 2.5 V vs SCE seem beyond the capabilities of $Sc(OTf)_3$ -catalysed oxidation. However, we observed that combination of high temperature (65 °C) and the addition of 0.2 equivalent of trifluoroacetic acid allows the oxidation of even such a demanding substrate as 4-(trifluoromethyl)toluene (1c) (see Fig. 3). It should be noted that stronger trifluoromethanesulfonic acid or the weaker acetic acid did not cause such a significant increase in oxidation conversion (see Supplementary Section S2 for details)."

c) The $[Sc(MeCN)_3(\eta^2-O_2)]^{3+}$ species is proposed to be the active catalyst in this transformation. However, the authors do not provide any direct evidence to support this claim.

We understand this remark. However, we would like to emphasize that the proposed complex is only an example of a possible active species. This example demonstrates in particular that oxygen coordination is important for changing the spectral properties. To further support the interaction of Sc(III) with oxygen, we performed additional experiments in the revised version. The coordination was studied using cyclic voltammetry, which is very sensitive to the interactions of redox-active species in solution. We have included the corresponding voltammogram (Fig. 5e) and discussion (see below) in the main text and also in Supplementary Section S9. Further support for the existence of the proposed species $[Sc(MeCN)_3(\eta^2-O_2)]^{3+}$ is an agreement of the results of the calculation of the excited state energies (Supplementary Section S12) with experimental results. For example, the theoretical fluorescence maximum (2.375 eV = 522 nm) agrees very well with the measured spectrum ($\lambda_{max} = 506$ nm). We have also included this discussion into the main article (see answer to question 4, reviewer 3).

Added text: "Interaction of Sc^{3+} ions with molecular O_2 was also supported by cyclic voltammetry experiments (Fig. 5e). Addition of $Sc(OTf)_3$ into oxygen-saturated acetonitrile was characterised by formation of a new peak ($E_{cp} = -0.95$ V vs SCE) with less negative potential compared to the peak of oxygen in acetonitrile ($E_{pc} = -1.16$ V vs SCE). The new signal is attributed to oxygen coordinated to Sc^{3+} ions. The same signal can be achieved by an inverse experiment. Cyclic voltammogram of $Sc(OTf)_3$ under argon contains peak at -1.83 V vs SCE corresponding to Sc^{3+} reduction. After bubbling the solution with oxygen gas, two peaks attributed to coordinated and "free" oxygen appeared (see Supplementary Section S9)."

Reviewer 3

This study presents the first report on the catalytic activity of Sc(OTf)₃ as a standalone photoredox catalyst, challenging the conventional perception that Sc³⁺ solely functions as a Lewis acid. This breakthrough pioneers a new direction for the application of rare-earth metals in photocatalysis. The work demonstrates the high efficiency and broad substrate compatibility of Sc(OTf)₃ in benzylic oxidation and direct aromatic ring cyanation, showcasing its promising practical potential. Mechanistic insights were rigorously validated through multiple approaches, including EPR spectroscopy, transient absorption spectroscopy, and theoretical calculations, significantly reinforcing the credibility of the conclusions.

The reaction was sophisticatedly designed and well conducted. The results reported in this paper are both significant and intriguing. This reviewer would like to recommend acceptance of this work after the following issues are addressed.

The authors are very grateful for very positive evaluation and appreciation of the pioneering nature of the work and our mechanistic studies. We also thank reviewer for the suggested additional studies that will contribute to further improvement of our manuscript.

1. The applicable scope of this reaction is limited, especially for the cyanation of arenes. In addition, regarding the low yield of substrates such as 4-(trifluoromethyl)toluene (1c), it is necessary to further discuss the relationship between its electronic structure and oxidation potential, or attempt to introduce co-catalysts to improve the performance.

The authors thank reviewer for this remark and realize that finding other possibilities to expand the substrate scope would be very interesting. In our experiments, we focused on oxidations and their support by protonation of a formed superoxide species. Indeed, it was shown that the addition of trifluoroacetic acid leads to the oxidation of even such a demanding substrate as 4-trifluoromethyltoluene ($E_{ox} = +2.61$ V vs SEC). At elevated temperature (65 °C) in propionitrile (instead of acetonitrile), this oxidation proceeded with a conversion of 33% to the corresponding benzoic acid. For the success of this oxidation, both Sc(OTf)₃ and trifluoroacetic acid are required, as demonstrated by independent experiments. The pK_a of the acid used is important, as shown by experiments with strong triflic acid or weaker acetic acid, which did not significantly support the oxidation. We have added the relevant discussion, including the discussion of the oxidation potentials of the substrates, to the main text and added the table with selected results as Table S4 into Supplementary Section S2. We also noticed this possible improvement at the end of mechanistic part.

Added text: "Substrates with higher oxidation potentials than 2.5 V vs SCE seem beyond the capabilities of Sc(OTf)₃-catalysed oxidation. However, we observed that combination of high temperature (65 °C) and the addition of 0.2 equivalent of trifluoroacetic acid allows the oxidation of even such a demanding substrate as 4-(trifluoromethyl)toluene (1c) (see Fig. 3). It should be noted that stronger trifluoromethanesulfonic acid or the weaker acetic acid did not cause such a significant increase in oxidation conversion (see Supplementary Section S2 for details)."

2. The author mentioned that O₂ acts as both an electron acceptor and a reactant. It is proposed to further verify the specific participation mode of oxygen through isotope-labeling experiments (such as using ¹⁸O₂).

Authors are really grateful for this suggestion. We performed model oxidation by using $^{18}\text{O}_2$ instead of $^{16}\text{O}_2$ oxygen and analysed the formed benzoic acid by ^1H NMR (for yield) and GC-HRMS (Plasmion SICRIT positive ionization) setup (for isotopic composition). The reaction mixture contained 76 % of benzoic acid and unreacted toluene. There was almost no benzoic acid $[\text{M}+\text{H}]^+ = 123.0441$ a.m.u. corresponding to acid with two ^{16}O atoms. On the other hand, majority of product (69 %) contained both ^{18}O atoms ($[\text{M}+\text{H}]^+ = 127.0525$ a.m.u.). Minor product (30 %) with $[\text{M}+\text{H}]^+ = 125.0483$ a.m.u. contained one ^{16}O and one ^{18}O . The fact that majority of benzoic acid contained two ^{18}O atoms shows that molecular oxygen is the source of oxygen atoms in benzoic acid. The minor product containing one unlabelled ^{16}O atom and one ^{18}O was explained by series of experiments described in detail in Supplementary Section S8. These experiments clearly show that the source of ^{16}O atom is water being present as a moisture in the reaction mixture and causes oxygen atom replacement in benzaldehyde intermediate (not benzoic acid) by oxo form-hydrate form equilibrium catalysed by Sc(III). Increased moisture is caused by oxophilicity of Sc(III). It was proved by independent experiment (amount of water in reaction mixture is approx. 3000 ppm when all additives are present which is in contrast of water content in used acetonitrile – 6-7 ppm). We did several affords to avoid increased moisture (also working in glove box) but we were not successful.

Text added: "Oxidation of toluene using isotopically labelled oxygen $^{18}\text{O}_2$ gave mainly benzoic acid (2a) with two incorporated ^{18}O atoms and almost no product containing two ^{16}O atoms (Fig. 5b). This result supports the hypothesis that molecular oxygen is the source of the oxygen atom in the product. Some amount (30 %) of product containing one unlabelled ^{16}O atom and one ^{18}O was also observed which is explained by enhanced residual moisture causing oxygen atom exchange in benzaldehyde intermediate by oxo form-hydrate form equilibria which is supported by Sc^{3+} (Lewis acid) catalysis (see Supplementary Section S8 for details)."

3. Supplement with the latest progress on "metal ion-coupled electron transfer (MCET)" in recent years (such as literatures after 2020) to reflect the timeliness of the research.

Authors thanks for this recommendation. We added recent literature [*JACS Au* **4**, 344-368 (2024); *ACS Omega* **9**, 38498-38505 (2024), *Adv. Energy Mater.* **10**, 1903933 (2020)] as references 25-27 into introductory part.

4. Add DFT calculations of the excited - state electronic structure to clarify the correlation between light absorption and catalytic activity.

Authors thanks for this suggestion. We calculated excited state energies including possible transactions. All calculated data (at DLPNO-CCSD(T) level of theory together with aug-cc-pVTZ basis and corresponding aug-cc-pVTZ/C auxiliary basis set in Orca software package) are shown in Supplementary Section S12. Reference was added into the main text. We also noticed that emission wavelength calculated for $[\text{Sc}(\text{MeCN})_3(\eta^2\text{-O}_2)]^{3+}$ (2.375 eV = 522 nm) corresponds to experimental spectrum ($\lambda_{\text{max}} = 506$ nm). Thus, we added this fact into the main text: "Also calculated emission (522 nm) corresponds to experimental fluorescence (around 506 nm; see Fig. 2d)."